# Applications of Platelet-Rich Fibrin (PRF) Membranes Alone or in Combination with Biomimetic Materials in Oral Regeneration: A Narrative Review

**DOI:** 10.3390/biomimetics10030172

**Published:** 2025-03-11

**Authors:** Javier Valenzuela-Mencia, Francisco Javier Manzano-Moreno

**Affiliations:** 1Department of Stomatology, Colegio Máximo de Cartuja s/n, University of Granada, 18071 Granada, Spain; valenzuelamencia@gmail.com; 2Biomedical Group (BIO277), University of Granada, 18071 Granada, Spain; 3Instituto Investigación Biosanitaria, ibs.Granada, 18012 Granada, Spain

**Keywords:** plasma, membranes, fibrin, biomimetic, bone regeneration

## Abstract

Platelet-rich fibrin (PRF) membranes are a biomaterial derived from the patient’s own blood, used in different medical and dental areas for their ability to promote healing, tissue regeneration, and reduce inflammation. They are obtained by centrifuging the blood, which separates the components and concentrates the platelets and growth factors in a fibrin matrix. This material is then moulded into a membrane that can be applied directly to tissues. The use of these PRF membranes is often associated with the use of different biomimetic materials such as deproteinized bovine bone mineral (DBBM), β-tricalcium phosphate (β-TCP), enamel matrix derivative (EMD), and hydroxyapatite (HA). Different indications of PRF membranes have been proposed, like alveolar ridge preservation, alveolar ridge augmentation, guided tissue regeneration (GTR), and sinus floor augmentation. The aim of this narrative review is to check the state-of-the-art and to analyze the existing gaps in the use of PRF membranes in combination with biomimetic materials in alveolar ridge preservation, alveolar ridge augmentation, guided tissue regeneration (GTR), and sinus floor augmentation.

## 1. Introduction

Platelet-rich plasma (PRP) is a platelet concentrate obtained from the patient’s own blood after centrifugation [1]. The use of PRP started in the early days of the field of haematology. Haematologists created this technique in the 1970s with the aim of creating a peripheral blood platelet concentrate so as to treat patients with thrombocytopenia [2]. Later, its use was extended to the field of bone regeneration in oral surgery and periodontics. It contains platelet concentrations 5–10 times higher than whole blood, proteins, protein-based bioactive factors, and leukocytes in varying amounts. The mechanism of action of PRP is based on the fact that the platelets contained in the autologous plasma concentrate release their alpha granules after the coagulation process has been activated locally at the wound site. These alpha granules contain a cocktail of growth factors that promote cell proliferation, chemotaxis, and differentiation, which are essential for osteogenesis. In addition to its procoagulant effect, PRP is a source of growth factors that are involved in the initiation and maintenance of wound healing by accelerating fibroblast repair and increasing tissue vascularisation. The addition of thrombin and calcium activates platelets to release proteins, cytokines, chemokines, and growth factors that are important for regulating cellular processes [1,3].

Activated platelet-derived factors serve as messengers and regulators that influence a variety of cell–cell and cell–extracellular matrix interactions and serve to modify the pericellular microenvironment [4,5,6].

Leukocyte- and platelet-rich fibrin (L-PRF) membranes consist of a fibrin matrix on which cells, growth factors, and certain cytokines are transported and released over time [7]. These cytokines are responsible for the activation and mitogenic response of the periosteum in bone regeneration. The use of L-PRF has been associated with the success of different treatments in periodontics and oral surgery that are related to bone repair and regeneration [8]. Fibrin membranes are biopolymers that can be used as scaffolds for the growth of mesenchymal stem cells and bone tissue cells [9]. The most important growth factors released by L-PRF membranes include vascular endothelial growth factor (VEGF), transforming growth factor beta (TGF-Beta), platelet-derived growth factor (PDGF), and fibroblast growth factor (FGF) [10,11]. PDGF release accelerates wound healing in hard and soft tissues, while TGF-Beta promotes connective tissue repair and bone regeneration [1].

L-PRF is a much more modern platelet concentrate, which is achieved with a simplified preparation without biochemical manipulation of the blood. This technique does not require anticoagulants, thrombin, or any other gelling agent. This is what characterises this concentrate, making it easily usable [12]. In addition, the new advanced PRF (A-PRF) protocol with a 14 min centrifugation time and 1500 rpm has been developed, which is expected to contain a relatively greater number of white blood cells (WBC) [13].

These platelet concentrates differ in their obtaining and processing, which influences their mode of presentation and final characteristics. Table 1 lists the main differences between these platelet concentrates [7,14,15,16].

Bone healing and regeneration is a complex process that is influenced by multiple factors. The most important steps in regeneration include haemostasis, inflammation, proliferation, and bone maturation and remodelling. It is essential to understand these principles in order to follow optimal and predictable clinical outcomes [17]. Today, many bone grafting biomimetic materials exist on the market that try to mimic the structure of the alveolar bone. These include xenografts (derived from other animal species and plants), synthetically fabricated alloplasts, and allografts that are harvested from human cadavers. Both have excellent osteoconductive properties, and certain types of allografts may also have osteoinductive properties through the release of bone morphogenetic proteins (BMPs) [18]. Each of these classes of bone grafts offers various advantages and disadvantages based on their respective handling properties, biocompatibility, surface geometry and chemistry, mechanical properties, and degradation properties. While autogenous bone is considered the gold standard, combining the features of osteoconduction, osteoinduction, and osteogenesis, alternative bone grafting materials that are available in higher supply with less patient morbidity have always been a desired end goal for clinicians [19].

Different indications of PRF membranes alone or in combination with biomimetic materials have been proposed, like alveolar ridge preservation [20], 1 and 2-stage sinus floor elevation [21], horizontal and vertical bone regeneration [22], regeneration of intrabony defects around teeth [23], the regeneration furcation defects [23], third molar extractions [20], and implant osseointegration [24]. PRF membranes also have adjunctive benefits in the treatment of medication-related osteonecrosis of the jaw [25] and endo surgery [26]. Platelet concentrates can be used alone or in combination with other materials. Its effects, when used in conjunction with other biomaterials, have been extensively studied in recent years [27,28,29,30].

Although many studies on the use of PRF have been published in the literature, there is a gap in the application of L-PRF membranes in different surgical techniques in the oral cavity since there is a great diversity of surgical protocols in which L-PRF is combined with a diversity of grafting materials, some of which are not standardised, which makes it difficult to compare the results from one study to another.

The aim of this narrative review is to check the state-of-the-art and to analyze the existing gap in the use of PRF membranes in combination with biomimetic materials in alveolar ridge preservation, alveolar ridge augmentation, guided tissue regeneration (GTR), and sinus floor augmentation.

**Table 1 biomimetics-10-00172-t001:** Comparison of different generations of platelet concentrates.

Type of Platelet Concentrate	Obtaining Procedure	Preparation	Characteristics
PRP (Platelet-Rich Plasma) [15]	Centrifugation of autologous blood.	1. Blood extraction. 2. First, centrifuge to separate red blood cells from plasma. 3. Second, centrifuge to concentrate platelets.	With a high platelet concentration and low leukocyte content, it is used in regenerative medicine and aesthetics.
PRF (Platelet-Rich Fibrin) [31]	Centrifugation of autologous blood without anticoagulants (usually 2700–3000 rpm for 12–14 min).	1. Blood extraction. 2. Immediate centrifugation without anticoagulants to form a fibrin matrix.	Natural fibrin matrix with a higher leukocyte content than PRP and a sustained release of growth factors.
L-PRF (Leukocyte- and Platelet-Rich Fibrin) [16]	Similar to PRF but with a specific lower centrifugation protocol (usually 2400–2700 rpm for 12 min).	1. Blood extraction. 2. Immediate low-speed centrifugation without anticoagulants to form a dense fibrin matrix.	Denser fibrin matrix that is rich in leukocytes and with a prolonged release of growth factors (7–14 days).
A-PRF (Advanced Platelet-Rich Fibrin) [15]	A variant of L-PRF with a modified centrifugation protocol (usually 1500–1800 rpm for 14–18 min).	1. Blood extraction. 2. Immediate lower speed and longer time centrifugation without anticoagulants.	More flexible fibrin matrix than L-PRF and with a greater release of growth factors.
i-PRF (Injectable Platelet-Rich Fibrin) [14]	A liquid variant of PRF obtained without anticoagulants. A centrifugation protocol of 700–1300 rpm for 3–5 min.	1. Blood extraction. 2. Immediate low-speed centrifugation to obtain a platelet-rich liquid.	The liquid form is ideal for injections.
PRGF (Plasma Rich in Growth Factors) [32]	Patented technique that differs from PRP in that it uses a specific anticoagulant and an activation protocol with calcium chloride.	1. Blood extraction. 2. Centrifugation with anticoagulant. 3. Activation with calcium chloride to release growth factors.	With a moderate platelet concentration and a low leukocyte content, it has a controlled release of growth factors.

## 2. Materials and Methods

A bibliographic search was carried out on the use and results obtained by PRF membranes and biomimetic materials in the main oral and maxillofacial surgical techniques where they are most frequently used, such as alveolar ridge preservation, alveolar ridge augmentation, guided tissue regeneration, and sinus floor augmentation.

An electronic search was carried out in the PubMed, Scopus, and Web of Science (WoS) databases. No time or language limitation was established.

The following combination of terms was used in the electronic search:

(platelet-rich fibrin) AND ((oral surgery) OR (bone regeneration) OR (alveolar ridge augmentation) OR (guided bone regeneration) OR (gbr) OR (ridge preservation) OR (alveolar preservation) OR (guided tissue regeneration) OR (dental implants) OR (sinus floor augmentation) OR (sinus lift).

Studies that met the following inclusion criteria were included as follows: randomised clinical trials (RCTs) conducted in humans with an L-PRF (including A-PRF) membranes collection protocol, as described by Dohan et al. [7]. We excluded studies that were not randomised clinical trials, those where PRF presentations other than membranes were used, and those that were not available in full text.

The included studies comprised a sample of systemically healthy patients over 18 years, excluding smokers, pregnant women, lactation with poor oral hygiene, periodontally compromised teeth, or taking any medication that could affect bone or soft tissue healing. The included studies had a follow-up of between 2 and 12 months.

The search and selection of articles were carried out by two independent researchers (F.-J.M.M. and J.V.-M.). The research results were reviewed to remove the duplicates, and the data from the studies included in this review were extracted independently in the same way by both researchers. Reference lists, including authors, titles, and abstracts, were screened to find relevant manuscripts. The bibliographies of eligible articles were also manually searched for missing papers after the electronic search.

## 3. Results and Discussion

### 3.1. L-PRF Membranes on Alveolar Ridge Preservation

Alveolar preservation is defined as any local therapeutic intervention performed in the socket immediately after extraction, aiming to maintain the hard and soft tissues as well as alveolar contours, usually with the intention of later implant placement in the area [33].

The filling of the alveolus in alveolar preservation has generally been performed using biomimetic materials such as xenografts, allografts, and alloplastic materials, with or without the use of membranes, as well as with products derived from the patient’s blood. Within the latter, platelet-rich fibrin (PRF) has been used either as the sole alveolar filling material or in combination with other biomimetic materials (Table 2) [34].

After tooth extraction, the disappearance of the fasciculated bone or bundle bone leads to a remodelling of the socket that results in bone resorption and, therefore, a loss of volume [35]. This is the healing procedure that follows the alveolus when no alveolar preservation procedure is performed. The aim of alveolar preservation is, therefore, to limit this post-exodontic bone remodelling in an attempt to maintain hard and soft tissue volumes as close as possible to the situation prior to tooth extraction [33].

When comparing performing alveolar preservation with PRF or allowing spontaneous healing of the post-extraction socket by means of a blood clot, the results obtained in the literature are controversial. Regarding horizontal and vertical bone dimensional changes, some studies find less bone resorption of the socket after alveolar preservation with PRF [34,36,37,38], while others find no change with respect to spontaneous healing [39,40,41,42,43]. When PRF has been used in combination with a collagen sponge, no differences have been obtained compared to PRF alone or spontaneous healing [44]. While most of these studies have measured volumetric bone changes by performing CBCT measurements, generally at depths of 1, 3, and 5 mm subcrestal [34,36,37,38], other studies have conducted these measurements by overlapping periapical radiographs or by measurements on the panoramic radiograph [34,36,37,38]. Some studies have also measured bone neoformation by histomorphometry [34,36,37,38]. By taking a bone sample using a trephine 2–3 months after alveolar ridge preservation, they have been able to measure the bone neoformation. While some studies found significantly higher neoformation percentages using PRF [34,36,37,38], other studies have found no significant difference [34,36,37,38].

The PRF procurement process used in all the alveolar preservation studies included in this review has followed the PRF procurement protocol described by Dohan et al. [7], where the blood obtained from the patient is centrifuged at 3000 rpm for 10′ or at 2700 rpm for 12′. However, there are other variations of PRF collection, such as A-PRF (advanced platelet-rich fibrin), where the centrifugation speed is decreased in order to increase the release of growth factors from the PRF membrane to, for example, to enhance fibroblast migration to the PRF membrane [45].

Castro et al. [42] compared L-PRF and A-PRF in alveolar preservation and found no significant differences in terms of a reduction in bone resorption after extraction, although they found similar bone neoformation in both cases and significantly more than spontaneous healing.

Another material frequently used in alveolar preservation is the allograft. Azangookhiavi et al. [46] compared alveolar preservation with mineralised particulate cortical allograft (FDBA), sealing of the socket with a free gingival graft, and filling of the socket with PRF and found no differences in terms of the changes in socket bone dimensions, concluding that the use of PRF in alveolar preservation shows acceptable efficacy as well as being low cost and easy to obtain.

**Table 2 biomimetics-10-00172-t002:** Application of PRF membranes on alveolar ridge preservation.

Authors, Year	Objectives	Treatment in Each Group	Results
Abad et al. 2023 [39]	To evaluate the efficacy of L-PRF in alveolar preservation compared to spontaneous healing.	-Control group: spontaneous healing. -Test group: L-PRF.	-Horizontal and vertical bone changes: similar horizontal bone width decreases in both groups. Lower vertical reduction in the test group (not significant).
Al Kassar et al. 2023 [34]	To radiographically evaluate the role of L-PRF in reducing post-extraction dimensional changes compared to spontaneous healing.	-Control group: spontaneous healing. -Test group: L-PRF.	-Vertical bone resorption of the vestibular plate was significantly lower in the test group (*p* = 0.004). Vertical bone resorption of the lingual plate was significantly lower in the test group (*p* = 0.032). Horizontal bone resorption: no differences between groups. Resorption in the width of the alveolar bone 6 mm subcrestal was significantly lower in the test group (*p* = 0.001).
Alasqah et al. 2024 [44]	To evaluate the effectiveness of PRF in maintaining alveolar dimensions after extraction, as well as its impact on post-extraction discomfort.	-Group I: PRF. -Group II: PRF + collagen. -Group III: spontaneous healing.	-Alveolar width: no significant differences between groups. Postoperative pain: higher levels of pain in the control group in the first 24 h. Similar levels of pain at the end of the first week.
Aravena et al. 2021 [40]	To determine the effectiveness of L-PRF versus spontaneous healing in alveolar preservation in clinical, radiographic, and volumetric terms.	-Control group: spontaneous healing. -Test group: L-PRF.	-Soft tissue healing: no significant differences between groups. Radiographic measurements: no significant differences between groups. Volumetric changes: no significant differences between groups.
Areewong et al. 2019 [41]	To compare the ratio of bone neoformation using PRF as alveolar preservation material with respect to spontaneous healing.	-Control group: spontaneous healing. -Test group: PRF.	-Bone neoformation: similar ratios between groups (31.33% in the test group vs. 26.33% in the control group).
Canellas et al. 2020 [36]	To evaluate the efficacy of L-PRF in alveolar preservation after tooth extraction.	-Control group: spontaneous healing. -Test group: L-PRF.	-Horizontal resorption was significantly lower in the test group at a depth of 1 mm and 3 mm from the bone crest (*p* = 0.0001). Vertical resorption of the vestibular plate was significantly lower in the test group (*p* = 0.028). Bone neoformation was significantly higher in the test group (*p* = 0.009).
Castro et al. 2021 [42]	To evaluate the dimensional changes in alveolar bone after tooth extraction when using L-PRF or A-PRF+ compared to spontaneous healing.	-L-PRF Group. -Group A-PRF+. -Spontaneous healing group.	-Horizontal vestibular and palatal resorption: no significant differences between groups. Vertical resorption: no significant differences between groups. Histological analysis: bone neoformation was similar between L-PRF and A-PRF+ and significantly higher than spontaneous healing (*p* < 0.05).
Mousavi et al. 2023 [43]	To assess the effects of L-PRF on alveolar changes following tooth extraction.	-Control group: spontaneous healing. -Test group: L-PRF.	-Alveolar width: Alveoli treated with L-PRF undergo significantly higher resorption in the most coronal portion of the alveolus. There were no significant differences between groups in terms of bone density, bone neoformation, or vertical changes in vestibular or lingual walls of the socket.
Sharma et al. 2020 [37]	To evaluate the clinical and radiographic influence of PRF on soft tissue healing and bone regeneration after tooth extraction.	-Control group: spontaneous healing. -Test group: PRF.	-Soft tissue healing: significantly better healing in the test group at 3 and 7 days (*p* = 0.025, *p* = 0.039). Bone regeneration: no significant differences between groups.
Temmerman et al. 2016 [38]	To investigate the influence of L-PRF as an alveolar filling material and its alveolar preservation properties.	-Control group: spontaneous healing. -Test group: L-PRF.	-Vertical vestibular resorption was significantly lower in the test group (*p* < 0.005). Horizontal resorption was significantly lower in the first lingual millimetre and in the first and third vestibular millimetres in the test group (*p* < 0.005). Total horizontal resorption: significantly lower horizontal resorption in the test group at 1, 3, and 5 mm (*p* < 0.005). Percentage of bone filling: significantly higher in the test group (*p* < 0.005). Sensation of postoperative pain: significantly lower on days 3, 4, and 5 in the test group.
Azangookhiavi et al. 2020 [46]	To compare the clinical application of allograft (FDBA) and PRF in alveolar preservation after extraction.	-Group A: FDBA + free gingival graft. -Group B: PRF.	-Changes in alveolar width and height: no significant differences between groups.

PRF: platelet-rich fibrin; L-PRF: leukocyte- and platelet-rich fibrin; FDBA: freeze-dried bone allograft.

### 3.2. L-PRF Membranes on Alveolar Ridge Augmentation or Guided Tissue Regeneration (GTR)

While there are a large number of studies published on the effects of PRF in alveolar preservation, studies examining PRF in alveolar ridge augmentation are minimal (Table 3). Hartlev et al. [47], in a randomised clinical trial, compared volumetric bone changes after completely autologous horizontal bone augmentation using an autologous bone block covered by PRF membranes or regeneration using biomimetic xenograft and collagen membranes. At 6 months after bone regeneration, no differences in bone volume changes were obtained between the two techniques.

Periodontal regeneration using different regeneration biomimetic materials, such as membranes, grafts, biologically active materials, or a combination of several of them, has shown significant clinical improvements in the treatment of intraosseous defects, with no clear differences between the different materials available [48].

Among the available techniques, flap surgery was one of the first techniques used [49], achieving successful results in the treatment of intraosseous defects [50].

Several papers have been included in this review that analyse whether the addition of PRF membranes to traditional flap surgery brings advantages to this procedure [51,52,53,54]. The combination of flap surgery and PRF has shown a significant improvement in the levels of probing depth, clinical attachment, and percentage of bone fill over the use of flap surgery alone. Bajaj et al. [49] found a significantly greater probing depth reduction (*p* < 0.05) and clinical attachment level gain (*p* = 0.003) by using PRF, and also a greater reduction of the intrabony defect depth (*p* < 0.001). Patel et al. [49] also found a significantly greater reduction of probing depth, clinical attachment level gain, and fill of the intrabony defect by using PRF (*p* = 0.001). The study by Pham et al. [54] also compared PRF and flap surgery with periodontal regeneration with a resorbable collagen membrane (OSSIX plus, Datum Dental, Seoul, Republic of Korea), with slightly superior results being achieved with the PRF. Pradeep et al. [53] also investigated the combination of hydroxyapatite and PRF. In this study, in terms of the probing depth and percentage of bone fill, both PRF alone and combined with hydroxyapatite obtained significant improvements compared to flap surgery alone, while in clinical attachment levels, only the combination of PRF and hydroxyapatite obtained a significant improvement.

Another widely used material in periodontal regeneration is an allograft. Naidu et al. [55] compared demineralised particulate allograft and its combination with PRF membranes and found no significant difference in terms of the percentage of bone fill, probing depth, or clinical attachment level between the two approaches.

Autologous bone and enamel matrix derivatives have also been frequently used in bone regeneration techniques. The study by Paolantonio et al. [56] evaluated the use of autologous bone in combination with amelogenins or PRF, finding improvements in both groups in terms of probing depth and clinical attachment but no differences between the groups.

**Table 3 biomimetics-10-00172-t003:** Application of PRF membranes on alveolar ridge augmentation or guided tissue regeneration (GTR).

Authors, Year	Objectives	Treatment in Each Group	Results
Hartlev et al. 2019 [47]	To evaluate the volumetric changes after horizontal regeneration using autologous bone covered by a PRF or xenograft membrane and a resorbable collagen membrane.	-Test group: autologous bone + PRF. -Control group: xenograft + collagen membrane.	-Bone volumetric changes: the bone resorption ratio was influenced by the region but not by treatment. No significant differences between groups.
Bajaj et al. 2017 [51]	To explore the efficacy of PRF in the treatment of intraosseous defects in aggressive periodontitis.	-Test group: flap + PRF. -Control group: flap.	-Significantly greater reduction in the probing depth and clinical insertion level in the PRF group (*p* = 0.05, *p* = 0.003). Bone filling of the defect was significantly higher in the PRF group (*p* < 0.001).
Naidu et al. 2024 [55]	To compare the clinical and radiographic effectiveness of DFDBA with PRF versus DFDBA alone in the treatment of intraosseous defects.	-Test group: DFDBA + PRF. -Control group: DFDBA.	-No significant differences were found between groups in terms of clinical and radiographic measures.
Paolantonio et al. 2019 [56]	To compare the combination of PRF and autologous bone with respect to the association of enamel matrix derivative (EMD) and autologous bone in the treatment of intraosseous defects.	-Test group: PRF + autologous bone. -Control group: EMD + autologous bone.	-All clinical and radiographic parameters improved significantly in both groups, with no differences between groups.
Patel et al. 2017 [52]	To assess the radiographic and clinical outcomes of PRF in intraosseous defects compared to flap surgery.	-Test group: flap + PRF. -Control group: flap.	-PRF significantly improves bone filling of the defect, as well as soft tissue healing and reduction of probing depth, compared to flap surgery.
Pham et al. 2021 [54]	To compare the clinical and radiographic outcomes of PRF, guided tissue regeneration, and flap surgery in the treatment of intraosseous defects.	-Group 1: flap + PRF. -Group 2: guided tissue regeneration. -Group 3: flap.	-At 12 months, significant improvements were found in all clinical and radiographic parameters in the 3 groups.
Pradeep et al. 2017 [53]	To explore the clinical and radiographic effectiveness of PRF versus hydroxyapatite with PRF in the treatment of intraosseous defects.	-PRF group: flap + PRF. -PRF + HA group: flap + PRF + hydroxyapatite. -Control group: flap.	-Greater reduction in drill depth in the PRF and PRF + HA groups compared to the control group. Greater clinical insertion gain in PRF and PRF + HA groups compared to the control group. A significantly higher percentage of bone filling was in the PRF and PRF + HA groups than in the control group.

PRF: platelet-rich fibrin; FDBA: freeze-dried bone allograft; DFDBA: demineralized freeze-dried bone allograft; EMD: enamel matrix derivative; HA: hydroxyapatite.

### 3.3. L-PRF Membranes on Sinus Augmentation

Different RCTs have been published in the literature on the use of L-PRF membranes in sinus floor augmentation (Table 4). Shiezadeb et al. [57] evaluated and compared the histomorphometric results obtained in sinus augmentation performed with allograft bone particles with or without PRF. The mean neoformed bone obtained in the PRF group was significantly higher than in the control group, concluding that the addition of L-PRF membranes to the allograft particles increases the amount of neoformed bone and decreases the amount of allograft particles remaining after a sinus lift. In the same sense, Pichotano et al. [58] compared the use of another biomimetic material, like deproteinized bone bovine mineral (DBBM) alone with DBBM + L-PRF membranes, for sinus augmentation cavity filling. The histological results showed a significant increase in the amount of neoformed bone in the L-PRF group, with less presence of residual bone particles, which would indicate that the use of L-PRF membranes would be useful for sinus augmentation surgery. In their RCT, Almeida-Malzoni et al. [59] evaluated the effect of the association of L-PRF with DBBM vs. DBBM alone in radiographic and histological sinus augmentation. The results showed that the combination of DBBM and L-PRF increased and accelerated new bone formation, allowing early implant placement, probably due to an increase in the expression of different growth factors such as runt-related transcription factor-2 (RUNX-2), vascular endothelial growth factor (VEGF), osteocalcin (OCN), and osteopontin (OPN).

In contrast, Cömert Kılıç et al. [60] compared in their RCT the use of beta-tricalcium phosphate alone or in combination with PRF membranes as a filling material in sinus augmentation. The histological results showed that a combination of PRF with beta-tricalcium phosphate did not increase the amount of neoformed bone in the sinus compared to beta-tricalcium phosphate alone. In the same sense, Nizam et al. [61], in their split-mouth RCT, also found no statistically significant differences at the histological level in the amount of neoformed bone at 6 months when comparing the use of DBBM vs. DBBM + L-PRF membranes in sinus augmentation. In another similar study, Dragonas et al. [62] compared the use of DBBM alone for sinus augmentation with the lateral window vs. DBBM + PRF membranes and found no statistically significant differences at the histological level between the two groups, concluding that the addition of PRF in the sinus augmentation techniques does not provide advantages with respect to neointimal bone gain.

Regarding the transcrestal sinus lift technique, Cho et al. [63] evaluated the survival rate, complications, and changes in residual alveolar bone height using saline or L-PRF membranes after a hydraulic transcrestal sinus lift. Filling with L-PRF produced a significant radiographic increase in neoformed bone compared to using saline alone. Along these lines, Lv et al. [64] evaluated the radiographic results of simultaneous implant placement with transcrestal sinus floor elevation (PESS) + L-PRF vs. lateral sinus floor elevation (LSFE) using DBMM. No statistical radiographic differences in marginal bone levels were found between the two techniques; however, postoperative morbidity was significantly lower in the PESS group.

## 4. Conclusions

It can be concluded that the use of PRF membranes seems to be beneficial in combination with different biomimetic materials for different oral regeneration techniques, such as alveolar ridge preservation, alveolar ridge augmentation, guided tissue regeneration (GTR), and sinus floor augmentation. However, more quality studies are needed to standardise the application technique of PRF membranes in order to achieve reliable results that are not so sensitive to the clinician. The molecular effects of PRF application in bone regeneration techniques need to be studied in more depth. Future studies should be performed focusing on the standardisation of the release of growth factors of the different PRF membranes and more RCT in order to finally determine the efficacy of PRF in oral regeneration techniques.

## Figures and Tables

**Table 4 biomimetics-10-00172-t004:** Application of PRF membranes on sinus augmentation.

Authors, Year	Objectives	Treatment in Each Group	Results
Shiezadeb et al. 2023 [57]	To evaluate and compare the histomorphometric outcomes of sinus floor elevation using allograft bone particles.	-Control group: Allograft bone particles. -Test group: Allograft bone particles + L-PRF.	-The mean amount of newly formed bone marrow was significantly higher in the test group (*p* = 0.044). -The average amount of remaining particles was also significantly less in the test group (*p* = 0.027).
Pichotano et al. 2019 [58]	To investigate the effectiveness of adding L-PRF to DBBM for early implant placement (4 months in the test group and 8 months in the control group) after maxillary sinus augmentation.	-Control group: DBBM. -Test group: DBBM + L-PRF.	-Histological evaluation demonstrated an increased percentage of newly formed bone for the test group compared to the control (*p* = 0.0087). The amount of residual graft in the control group was significantly higher than in the test group (*p* = 0.0003).
Almeida-Malzoni et al. 2023 [59]	To evaluate the effect of the association between L-PRF and DBBM in maxillary sinus augmentation.	-Control group: DBBM and implant placement after 8 months. -Test group: DBBM + L-PRF and implant placement after two different periods (4 and 8 months).	-Histologically, the test group showed a significant increase in bone neoformation compared to the control group and a lesser percentage of residual graft from T1 (1 week after surgery) to T2 (before implant placement).
Cömert Kılıç et al. (2017) [60]	To compare the histologic and histomorphometric outcomes of maxillary sinus floor augmentation among β-TCP alone, P-PRP-mixed β-TCP, and PRF-mixed β-TCP.	-β-TCP alone. -P-PRP-mixed β-TCP. -PRF-mixed β-TCP.	-No significant differences in new bone formation or residual graft particles were found between groups.
Nizam et al. 2017 [61]	To evaluate the effect of L-PRF in combination with DBBM on bone regeneration in maxillary sinus augmentation.	-Control group: DBBM. -Test group: DBBM + L-PRF.	-There was no qualitative difference in histological analyses among the groups.
Dragonas et al. 2023 [62]	To analyse and compare the effects of advanced platelet-rich fibrin (A-PRF) and plasma rich in growth factors (PRGF) combined with deproteinized bovine bone mineral (DBBM) on bone regeneration outcomes in maxillary sinus augmentation (MSA) procedures.	-Control group: DBBM. -Test group 1: DBBM + A-PRF. -Test group 2: DBBM + PRGF.	-No significant differences in new bone formation or residual graft particles were found between groups.
Cho et al. 2020 [63]	To evaluate the implant survival rate, any complications, and changes in residual alveolar bone height using saline or PRF filling after hydraulic transcrestal sinus lifting.	-Control group: Saline. -Test group: PRF.	-PRF filling induced significantly more radiographic bone gain compared with saline (*p* < 0.05).
Lv et al. 2022 [64]	To evaluate patient-reported outcomes and radiographic results of simultaneous implant placement in severely atrophic maxilla using flapless endoscope-assisted osteotome sinus floor elevation (PESS) with PRF and to compare the results with those of lateral sinus floor elevation (LSFE).	-PESS + PRF group. -LSFE + DBBM.	-No statistical radiographic differences in marginal bone levels were found between the two groups; however, postoperative morbidity was significantly lower in the PEES + PRF group.

PRF: platelet-rich fibrin; L-PRF: leukocyte- and platelet-rich fibrin; PRGF: plasma rich in growth factors; DBBM: deproteinized bovine bone mineral; β-TCP: β-tricalcium phosphate; MSA: maxillary sinus augmentation; PESS: osteotome sinus floor elevation; LSFE: lateral sinus floor elevation.

## Data Availability

The data presented in this study are available on request from the corresponding author.

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
