# Peer review of "Applications of Platelet-Rich Fibrin (PRF) Membranes Alone or in Combination with Biomimetic Materials in Oral Regeneration: A Narrative Review"

_biomimetics, 2025, doi:10.3390/biomimetics10030172_

Round 1

Reviewer 1 Report

Comments and Suggestions for Authors

The research investigates the applications of platelet-rich fibrin (PRF) membranes alone or in combination with biomimetic materials in oral regeneration. Specifically, it examines their effectiveness in alveolar ridge preservation, guided bone regeneration (GBR), guided tissue regeneration (GTR), and sinus floor augmentation. The review assesses the current state of research, comparing PRF with other regenerative techniques.

The topic is highly relevant and addresses a significant gap in the field of oral and maxillofacial regeneration. While PRF has been extensively studied, its comparative effectiveness with biomimetic materials remains controversial. This review attempts to consolidate findings from various randomized clinical trials (RCTs) to determine whether PRF provides a significant advantage over conventional techniques. Given the increasing emphasis on biological and minimally invasive regenerative approaches, the study is both original and important.

However, some concerns are:

  • The inclusion/exclusion criteria could be more stringent, particularly in defining PRF protocols (e.g., variations like L-PRF, A-PRF).
  • Standardized outcome measures (e.g., bone volumetric changes, histological assessments) should be consistently reported across studies.
  • The role of confounding factors (e.g., patient health status, surgical techniques, follow-up duration) should be discussed more thoroughly.
  • The study acknowledges that PRF shows potential regenerative effects, but the evidence remains controversial due to variations in study design and methodology. The authors correctly call for more high-quality RCTs and standardization of PRF protocols. However, the conclusion could be stronger in highlighting specific clinical recommendations (e.g., in which situations PRF should be preferred).
  • Seems unfinished sentence on the Table 3. Conclusion of Cömert Kılıç et al. (2017) [48] .

Reviewer 2 Report

Comments and Suggestions for Authors

A narrative review in title is recommended.

Table 2. 

Ref. #39 Bajaj et al, Ref.#40, Patel et al, Ref. #42 Pham et al, Ref. #41 Pradeep et al.GBR vs GTR: the defect characteristic of extraction socket and perio defect somewhat different, so that the comparison in this review may not be inappropriate.

Table 3.

 Ref #46, Pichotano: when is the early implant placement after 8 months of healing in sinus?  

Ref #47, please provide evaluation time and early implant placement time.

Ref. #51. Cho et al. is inappropriate to review here due to experimental design, and conclusion.

Ref #52. Lv et al. is inappropriate to review here because PRF and DBBM were put into sinus with different surgical approaches. Recommend comparison PRF and DBBM in a same procedure. In addition the conclusion commented the superiority of surgical procedure not PRF and DBBM.

Reviewer 3 Report

Comments and Suggestions for Authors

The type of review should be mentioned in the title. Eg- narrative. In the abstract, alter the sentence in line 21-22 regarding aim. As the review is on oral regeneration it would be better to specify whether it is in the field of periodontics or oral and maxillofacial surgery. The introduction could be improved by mentioning the principles and factors regarding wound healing and regeneration. The concepts of developement of various platelet concentrates and its history.

A tabular column explaining the different generations of platelet concentrates and how they are prepared can be added. The advantage of combining with biometric materials can be mentioned. Emphasis to be given to the particular PRF that is being discussed.

Reviewer 4 Report

Comments and Suggestions for Authors

Dear Authors,

Thank you for your efforts in writing this narrative review. Before we consider your manuscript for publication, please address the following questions and suggestions for improving the manuscript.

  1. Line 57 - the authors have not mentioned allografts, which are derived from the same species (human) but with a different genotype. Please incorporate basic information regarding the use of allografts in the introduction section. This addition would provide a more comprehensive overview of graft types and their applications in regenerative medicine.
  2. Before defining the aim of this narrative review please clearly state the existing gap in the literature that this review aims to address. What specific area of knowledge or understanding is currently lacking in the field? Explain why you chose to conduct a narrative review rather than a systematic review. What advantages does this approach offer for addressing the identified research gap? How does it align with the objectives of your study?please state the research gap, and why you have decide to do a narrative review approach instead of the systematic review?
  3. The Results and Discussion sections require detailed revision. The surgical indications described in subsections 3.1 – 3.3 are not correctly terminologically written. Alveolar ridge preservation (socket preservation), sinus floor augmentation are essentially variations of Guided Bone Regeneration (GBR)/Guided Tissue Regeneration (GTR) procedures. The fundamental aspect of these procedures is the use of resorbable or non-resorbable membranes. For instance, studies like Nizam et al. (49) and Pichotano et al. (46) utilized resorbable collagen membranes. Also, the current table presentation is more typical of systematic/scoping reviews, which does not align with a narrative review approach. If tables are retained, they should be formatted in landscape mode to enhance readability. A comprehensive reconstruction of the Results and Discussion section is necessary to accurately represent the interconnected nature of these regenerative procedures and their underlying principles.
  4. Overall, I see potential in this paper; however, the concerns outlined above should be carefully addressed for better scientific quality and clarity. The strength of this review lies in the inclusion of recent studies published in high-impact journals, which adds significant value to the manuscript. I look forward to your response and the revised version of the manuscript!

Round 2

Reviewer 4 Report

Comments and Suggestions for Authors

Dear Authors, thank you for addressing all of my comments. 

Wish you all the best, The Reviewer